# Waldenstrom Macroglobulinemia Recurrence with Bing–Neel Syndrome Presentation

**DOI:** 10.3390/reports7020034

**Published:** 2024-05-07

**Authors:** Raffaella Capasso, Miriam Buonincontro, Ferdinando Caranci, Antonio Pinto

**Affiliations:** 1Department of Radiology, CTO Hospital, Azienda Ospedaliera dei Colli, Viale Colli Aminei 21, 80141 Naples, Italy; antonio.pinto@ospedalideicolli.it; 2Department of Pulmonology and Oncology, Respiratory Intensive Care Unit, Monaldi Hospital, 80131 Naples, Italy; miriam.buonincontro@ospedalideicolli.it; 3Department of Precision Medicine, University of Campania Luigi Vanvitelli, 80134 Naples, Italy; ferdinando.caranci@unicampania.it

**Keywords:** Bing–Neel syndrome, Waldenstrom macroglobulinemia, leptomeningeal enhancement, monoclonal bands, ibrutinib

## Abstract

Bing–Neel syndrome (BNS) is a rare condition that may occur in patients with Waldenstrom macroglobulinemia (WM) and is caused by lymphoplasmacytic infiltration into the central nervous system. BNS is an extramedullary manifestation of WM which may present with various neurological signs and symptoms that make the diagnosis difficult to achieve. We present a case of BNS in a 60-year-old patient diagnosed 6 years after recovering from Waldenstrom’s macroglobulinemia. We observed the patient for a secondary generalized focal motor seizure. Unenhanced brain CT revealed slight hyperdensity of left parietal subarachnoid spaces. The MRI of the brain and spinal cord showed leptomeningeal enhancement in both parietal lobes. The presence of monoclonal bands (light chain k and IgM) was found in cerebrospinal fluid, leading to the diagnosis of BNS. The patient started treatment with ibrutinib and remains clinically stable during a 1-year follow-up. However, the MRI showed the appearance of a new subcortical left parietal lesion. BNS is an extremely rare presentation of WM that should be recognized and considered early in the presence of unexplained neurological symptoms in patients with a history of WM, even if the patient appears to have recovered.

## 1. Introduction

Waldenstrom’s macroglobulinemia (WM) is a lymphoproliferative B cell disorder characterized by the production of monoclonal immunoglobulin (Ig) M. Bone marrow and lymphoid tissue are infiltrated by lymphoplasmacytic cells and plasma cells. Peripheral neurological involvement can occur in up to 47% of patients [1,2,3,4]. Central nervous system (CNS) lymphoplasmatic infiltration in WM is a rare, poorly defined, and probably underdiagnosed manifestation of WM known as Bing–Neel syndrome (BNS) [1,2]. BNS affects approximately 1% of patients with WM [5]. 

BNS presents a difficult diagnosis due to the rarity of the syndrome and the variety of clinical signs and symptoms. In addition, some symptoms of WM may be related to symptoms observed in other complications like hyperviscosity syndrome or anti-myelin associated glycoprotein (MAG) antibody-related neuropathy [6,7]. BNS may occur at any stage of WM disease, which makes diagnosis even more difficult. BNS can present itself in patients with known WM, even when there is no systemic progression, even up to 25 years after the diagnosis of WM, as well as in patients who have never been diagnosed [8,9,10].

We report a case of BNS diagnosed 6 years after the patient had recovered from WM.

## 2. Detailed Case Description

A 60-year-old Caucasian male presented with a secondary generalized focal motor seizure that caused post-ictal right hyposthenia, which resolved after 2 days. Neurological examination showed a slight leveling of the left lower limb during the Mingazzini maneuver, hyperelicitable osteotendinous reflexes in the four limbs with anisoreflexia left > right, pathological Babinski’s sign on the right, and hypopallesthesia of the lower limbs prevalent on the right with a cranio-caudal gradient. The patient had suffered from Waldenstrom macroglobulinemia 6 years before, but he recovered after treatment with Rituximab and Bendamustine. The patient did not have a history of epilepsy or any recent history of fever, vaccine, upper respiratory tract infection, abdominal symptoms, trauma, or travel. He suffered from psoriatic arthritis in treatment while taking Apremilast (300 mg × 2/day). At the time of presentation, the routine blood tests were unremarkable except for a slight increase in erythrocyte sedimentation rate (21 mm/h), lymphocytopenia (0.9 × 10^2^/μL), and thrombocytopenia (127 × 10^3^ μL). An unenhanced brain CT scan at the clinical onset revealed faint hyperdensity in some left parietal subarachnoid spaces (SHSs), which replaced the physiological hypodensity of the liquor (Figure 1). Patients underwent MRI scan after 5 days that revealed pathological signal and leptomeningeal enhancement on both left and right parietal SHSs without diffusion restriction (Figure 2). Further MRI examination performed one week later was negative for spinal cord and radicular/cauda equina alterations (Figure 3) but confirmed the stability of leptomeningeal findings. Leptomeningeal contrast enhancement was suspected for leptomeningitis, post-ictal hyperemia, and metastasis. After 20 days, the patient also underwent a total body CT scan to exclude any primary malignancies. Peripheral blood tumor marker (CEA, aFP, CA19-9, CA125, TPA, PSA, NSE, CYFRA), infective (HSV, HZV, JVC, EBV), and autoimmunity (anti-dsDNA, anti-RNP, anti-Sm) tests were negative. More than one month after the onset, cerebrospinal fluid (CSF) analysis documented the presence of monoclonal bands (light chain k, IgM) with elevated IgM index (0.45) calculated by [CSF IgM (mg/L)/serum IgM (g/L)]/[CSF albumin (mg/L)/serum albumin (g/L)] that is [1.89 (mg/L)/7.13 (g/L)]/[28.3 (mg/L)/48 (g/L)]. CSF flow cytometry revealed 2527 cells and 850 lymphocytes (33.64%). Among CSF lymphocytes, 87% of them were CD3+ CD5 +, while only 8% of them were CD19+ CD20+. The CSF tests did not reveal the presence of infectious meningitis (HSV1, HSV1, EBV, Adenovirus, Enterovirus, Parechovirus). The polymerase chain reaction test was negative for the L265P mutation in the MYD88 gene in both CSF and blood samples. CSF and MRI findings, along with the history of WM, were used to suggest the diagnosis of BNS within 2 months of the clinical onset (diagnostic timeline, Figure 1). Therefore, brain tissue biopsy was not performed. A treatment with ibrutinib (140 mg three times daily) was initiated for the patient and is still receiving it. During a 1-year follow-up, there have been no new neurological signs or symptoms. The latest brain MRI showed a new subcortical left parietal lesion (18 mm) that was suspected to be caused by neoplastic cells infiltrating white matter vessels. Despite this, the spectroscopy map did not reveal any abnormal metabolic peaks in the lesion (Figure 4). The patient accepted the diagnosis and started and followed the therapy correctly. Despite his thrombocytopenia worsening, he tolerated the treatment well.

## 3. Discussion

The meninges and vessels are infiltrated by B cells, leading to the accumulation of plasma cells in Virchow–Robin spaces, as characterized by BNS on histopathology. According to this explanation, BNS is most often found in meningeal sheets and subependymal/periventricular regions within the CNS, while subcortical regions or the brainstem are not often affected [2,10]. In cases of suspected BNS, MRI of the brain and spinal cord is both recommended and necessary. Neuroimaging is beneficial in supporting BNS diagnosis, excluding differential diagnoses (primary neoplasm, infectious, and others), and identifying a potential biopsy site. While MRI is better at detecting meningeal pathology, our patient’s initial suspicion of leptomeningeal disease was based on the CT scan. MRI is the imaging modality of choice for diagnosing BNS. Fluid-attenuated inversion recovery (FLAIR) and T1-weighted sequences prior to and following gadolinium administration provide the most accurate sequences. In order to prevent mass effects, obstructive hydrocephalus, and non-specific meningeal contrast enhancement, it is suggested to undergo MRI prior to lumbar puncture [11]. On MRI, CNS involvement can be distinguished into diffuse or tumoral forms. Leptomeningeal structures and periventricular white matter are the main targets of tumoral cells in the most common diffuse infiltrative form. The less common tumoral forms, unifocal or multifocal, tend to be situated in the deep subcortical regions of the brain. Contrast enhancement and/or thickening of meningeal sheaths can indicate CNS infiltrations on CT or MRI, which can involve brain tissue and/or the nerve root of the cauda equina [7,9]. CT and MRI findings were consistent with the diffuse form of BNS in our case. Accurate diagnosis of BNS can be made via biopsy and cerebrospinal fluid examination. In most cases, such as our patient, biopsy is more difficult to perform without neurological impairment, and it is more invasive than lumbar puncture [7]. Lumbar puncture plays an essential role in BNS diagnosis since it identifies CSF leukocytosis and allows for flow cytometric analysis and/or molecular studies on B cells. It is important to analyze the CSF sample as soon as it is collected. Although the M-component could be detected through electrophoresis on the CSF sample, it is not usually performed in routine practice [8]. CSF electrophoresis was helpful in confirming the diagnosis of BNS in our patient, along with the excessive B cell presence in the CSF. The elevated IgM index (normal reference range <0.060) unequivocally indicated that the concentration of IgM monoclonal protein in CSF far above what would have been expected from passive leakage across the brain–blood barrier [11,12]. Positive expression of pan B cell markers (such as C19 and CD20) is a characteristic of the CSF flow cytometric profile of cells in BNS, as shown in our patient [13]. It is unusual that in our patient a CD5 + prevalence was also observed, despite WM being typically CD5-. The mutation of the myeloid differentiation primary response 88 (MYD88 L256P) gene has been recently described as a useful diagnostic parameter for BNS. Our patient did not have this mutation, even though it is observed in the vast majority of patients with WM [5]. In our case, laboratory and imaging findings ruled out the other suspected causes of leptomeningeal enhancement such as leptomeningitis, post-ictal hyperemia, and metastases. Kappa-type light chain detected on the CSF immunofixation electrophoresis, bilateral leptomeningeal sheet involvement, and a history of WM led to BNS diagnosis for our patient. As a limitation of this report, our patient did not perform bone marrow or brain biopsy. Until now, there has been no clear agreement on how to treat BNS. The ability of Bruton tyrosine kinase inhibitors, such as ibrutinib, to pass the blood–brain barrier makes it a promising treatment option [7]. According to recent evidence, it is recommended to start treatment with a dose of 420 mg and, if there is no response, to increase it to 560 mg [14]. The most frequent side effect reported in the literature for ibrutinib is lymphopenia (19%). Additionally, ibrutinib therapy is associated with an increase in infection rates, thrombocytopenia, and transaminase levels [15]. Despite the appearance of a new lesion on brain MRI, our patient has remained clinically stable under ibrutinib treatment for almost a year. To assess the evolution of BNS and the safety of the treatment, a longer-term follow-up is necessary. 

## 4. Conclusions

In conclusion, BNS is an extremely rare presentation of WM that should be recognized and considered early in the presence of unexplained neurological symptoms in patients with a history of WM, even if the patient appears to have improved.

## Data Availability

The raw data supporting the conclusions of this article will be made available by the authors, without undue reservation.

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
