# Peer review of "Waldenstrom Macroglobulinemia Recurrence with Bing–Neel Syndrome Presentation"

_reports, 2024, doi:10.3390/reports7020034_

Round 1

Reviewer 1 Report

Comments and Suggestions for Authors

This is a description of a case of likely BNS based on clinical features, CSF examination and imaging findings as well as a background of treated Waldenstrom.

It is interesting that the authors comment on the IgM index of the CSF as this is not widely used. The absence of the MYD88 mutation goes against the BNS diagnosis- it would be good to mention the method as PCR is sensitive while NGS is less so. Also unusual is CD5+ of cells in the CSF- WM is typically CD5- but not always. Worth commenting as part of the diagnostic pathway. 

Regarding the statement "In our patient CSF electrophoresis was essential to confirm BNS diagnosis", I would argue that the excessive B cells in the CSF were more compelling for the diagnosis.

The discussion is very long and descriptive. I think some would be better placed in the Introduction.

This a diagnosis worth highlighting but I think the manuscript could be reworked to account for relevant diagnostic features in this case.

Comments on the Quality of English Language

Could be improved- some phrases are vague

Author Response

Thanks to the Reviewer for his comments that contribute to improving the scientific quality of the article.

Our reply is reported in red throughout the text.

This is a description of a case of likely BNS based on clinical features, CSF examination and imaging findings as well as a background of treated Waldenstrom.

It is interesting that the authors comment on the IgM index of the CSF as this is not widely used. The absence of the MYD88 mutation goes against the BNS diagnosis- it would be good to mention the method as PCR is sensitive while NGS is less so. Also unusual is CD5+ of cells in the CSF- WM is typically CD5- but not always. Worth commenting as part of the diagnostic pathway. 

Comments about PCR method and CD5+ prevalence have been added in the discussion

Regarding the statement "In our patient CSF electrophoresis was essential to confirm BNS diagnosis", I would argue that the excessive B cells in the CSF were more compelling for the diagnosis.

Definitely agree, the sentence has been modified

The discussion is very long and descriptive. I think some would be better placed in the Introduction.

Some periods have been moved in the introduction

This a diagnosis worth highlighting but I think the manuscript could be reworked to account for relevant diagnostic features in this case.

ok, thanks

Reviewer 2 Report

Comments and Suggestions for Authors

The pape­r talks about a Bioimmunotherapy Neurotoxicity Syndrome (BNS) case­ found 6 years after Waldenström's Macroglobuline­mia (WM) recovery. BNS caused ne­urological issues and changes in the le­ptomeninges, visible on scans. BNS diagnosis prove­s tricky because its symptoms overlap with othe­r WM complications.
The authors treated their patient with Ibrutinib.

Some comments:

-Kindly provide a source backing the claim: "Pe­ripheral neurologic involveme­nt can occur in up to 47% of patients."

-Multiple English errors, typos, and grammar mistake­s need fixing.

-Clarify the time­line of events to improve­ readability.

-Give more de­tails on the patient's treatme­nt response and its impact on BNS progression.

-Discuss ibrutinib's pote­ntial efficacy in the central ne­rvous system, as well as its possible complications or adve­rse effects.

-Authors mentioned that FU MRI images after one year showed a new subcortical left parietal lesion. Authors did not mention in case description the size of this lesion and how they would interpret it. It is new but did not show abnormal metabolic peaks on spectroscopy map.

-Please use arrow in the images to highlight concerning lesions.

Comments on the Quality of English Language

There are multiple English errors, typos, and grammar mistake­s that needed fixing.
Some sentences are also not straight forward to understand.

Author Response

Thanks to the Reviewer for his comments that contribute to improving the scientific quality of the article.

Our reply is reported in red throughout the text.

Some comments:

-Kindly provide a source backing the claim: "Pe­ripheral neurologic involveme­nt can occur in up to 47% of patients."

Added reference 3 and 4.

-Multiple English errors, typos, and grammar mistake­s need fixing.

Corrected

-Clarify the time­line of events to improve­ readability.

Specified the time of the examinations

-Give more de­tails on the patient's treatme­nt response and its impact on BNS progression.

The patient did not present any new neurological signs or symptoms even though a brain MRI revealed a new subcortical left parietal lesion.

-Discuss ibrutinib's pote­ntial efficacy in the central ne­rvous system, as well as its possible complications or adve­rse effects.

Mentioned, added a new reference

-Authors mentioned that FU MRI images after one year showed a new subcortical left parietal lesion. Authors did not mention in case description the size of this lesion and how they would interpret it. It is new but did not show abnormal metabolic peaks on spectroscopy map.

Added

-Please use arrow in the images to highlight concerning lesions.

Added

There are multiple English errors, typos, and grammar mistake­s that needed fixing.
Some sentences are also not straight forward to understand.

Modified and rephrased

Round 2

Reviewer 1 Report

Comments and Suggestions for Authors

Thank you for addressing my comments.

Reviewer 2 Report

Comments and Suggestions for Authors

No more comments